# Input–output maps are strongly biased towards simple outputs

Kamaludin Dingle[1,2,3], Chico Q. Camargo[1,2] & Ard A. Louis[1]

Many systems in nature can be described using discrete input–output maps. Without knowing details about a map, there may seem to be no a priori reason to expect that a randomly chosen input would be more likely to generate one output over another. Here, by extending fundamental results from algorithmic information theory, we show instead that for many real-world maps, the a priori probability $P(x)$ that randomly sampled inputs generate a particular output $x$ decays exponentially with the approximate Kolmogorov complexity $\tilde{K}(x)$ of that output. These input–output maps are biased towards simplicity. We derive an upper bound $P(x) \lesssim 2^{-a\tilde{K}(x)-b}$, which is tight for most inputs. The constants $a$ and $b$, as well as many properties of $P(x)$, can be predicted with minimal knowledge of the map. We explore this strong bias towards simple outputs in systems ranging from the folding of RNA secondary structures to systems of coupled ordinary differential equations to a stochastic financial trading model.

[1] Rudolf Peierls Centre for Theoretical Physics, University of Oxford, Oxford, OX1 3NP, UK. [2] Systems Biology DTC, University of Oxford, Oxford, OX1 3QU, UK. [3] International Centre for Applied Mathematics and Computational Bioengineering, Department of Mathematics and Natural Sciences, Gulf University for Science and Technology, P.O. Box 7207, Hawally 32093, Mubarak Al-Abdullah, Kuwait. Correspondence and requests for materials should be addressed to K.D. (email: dingle.k@gust.edu.kw) or to A.A.L. (email: ard.louis@physics.ox.ac.uk)

Discrete input–output maps are widely used in science and engineering. Many maps are intrinsically discrete, such as models of the mapping from genotypes to discrete phenotypes in biology, or networks of Boolean logic functions in computer science. But discrete maps also arise naturally by coarse-graining continuous systems. Examples include differential equations, where the inputs are discretised values of the equation parameters, and the outputs are discretised values of the solutions for a given set of boundary conditions. Such a wide diversity of map types might at first sight suggest that, without knowing details of a particular map, there are no grounds for predicting one output to be more likely than another.

On the other hand, a closely related problem has been studied, albeit in an abstract way, in a field called algorithmic information theory (AIT), founded by Solomonoff[1,2], Kolmogorov[3] and Chaitin[4,5]. Central concepts in AIT include the universal Turing machine (UTM), an abstract computing device that can compute any function[6], and the Kolmogorov–Chaitin complexity or simply Kolmogorov complexity $K_U(x)$ of a binary string $x$, defined as the length of the shortest input program $p$ that generates output $x$ when it is fed into a prefix UTM $U$. Technically the Kolmogorov complexity is always defined with respect to a particular UTM, but this is often ignored because of the invariance theorem[7,8] which states that if $K_U(x)$ and $K_V(x)$ are the Kolmogorov complexities defined w.r.t UTMs $U$ and $V$, respectively, then we can write $K_U(x) = K_V(x) + \mathcal{O}(1)$, where $\mathcal{O}(1)$ denotes terms that are asymptotically independent of $x$ (or equivalently, $|K_U(x) - K_V(x)| \leq M_{U,V}$, where $M_{U,V}$ is a constant independent of $x$). In the limit of large complexities these $\mathcal{O}(1)$ differences can be neglected and one speaks simply of the Kolmogorov complexity $K(x)$ which is a property of $x$ only. We provide a short pedagogical description of AIT pertinent to the current paper in Supplementary Note 1. More complete accounts can be found in standard textbooks[7,8].

Historically, the first formulations of AIT, by Solomonoff[1,2], arose from studying the probability $P_U(x)$ that random input programs fed into a UTM $U$ generate output $x$. For technical reasons, it is easiest to consider UTMs that only accept prefix codes[7] for which no program is a prefix of another. For such codes, the probability that a random binary input program of length $l$ is chosen is $2^{-l}$. The most likely input that generates $x$ is then the shortest string to do so: by definition, a string of length $K_U(x)$. Since there can also be longer inputs that generate $x$, this means there is a lower bound $2^{-K_U(x)} \leq P_U(x)$. Later Levin[9] also proved an upper bound in what is now called the AIT coding theorem:

$$2^{-K(x)} \leq P(x) \leq 2^{-K(x)+\mathcal{O}(1)}, \qquad (1)$$

where we have dropped the subscript $U$ due to the invariance theorem. The upper bound in the coding theorem is neither obvious or straightforward. Intuitively, this fundamental result means that 'simple' outputs, with smaller $K(x)$, have an exponentially higher probability of being generated by random input programmes for a UTM than complex outputs with larger $K(x)$. This prediction is completely at odds with the naive expectation that all outputs are equally likely.

While these results from AIT are general and elegant, their direct application to many practical systems in science or engineering suffers, unfortunately, from a number of well-known limitations. First, due to the halting problem[6], $K(x)$ is formally uncomputable, meaning that there cannot exist any general method that takes $x$ and computes $K(x)$[7]. Second, many key AIT results, such as the invariance theorem or the coding theorem, only hold up to $\mathcal{O}(1)$ terms which are unknown, and therefore can only be proven to be negligible in the asymptotic limit of

large $K(x)$ values, while real-world applications frequently concern systems that are not in the asymptotic limit. Third, many input–output maps from science or engineering are computable, that is all their inputs can be mapped to outputs so that they have no halting problem and are not UTMs. Therefore many results from AIT, which typically rely on special properties of UTMs, may not be directly applicable.

On the other hand, the basic intuition behind the coding theorem—complex outputs are harder to generate by random sampling of inputs than simpler ones are—may be quite general. Moreover, the coding theorem prediction is very strong: an exponential decrease in probability upon a linear increase in complexity. Such a strong relationship might be expected to have influence even in situations where not all the conditions for its derivation within an AIT context are met. These intuitions beg the question of how the coding theorem, or closely related concepts, can help make predictions about probabilities in concrete real-world input–output systems.

In the next sections, we derive a weaker version of the coding theorem, which approximately preserves the exponential preference for low complexity. In particular, this allows us to make practical predictions for a broad class of maps. We explicitly demonstrate the exponential bias towards low-complexity outputs in systems ranging from RNA folding to coupled ordinary differential equations to a financial trading model.

## Results

**Coding theorem for computable functions.** With these questions about practical applications in mind, we consider computable maps of the form $f{:}I \to O$, where $I$ is a collection of input sequences, and $O$ is the corresponding collection of outputs, which can also be described (either because they are discrete objects, or by coarse-graining) as discrete sequences. We denote the size of the input space $I$ of the map by $n$, e.g., for binary sequences of fixed length $n$ the size is $2^n$ possible inputs.

Following a standard procedure from AIT[7,10], each output $x \in O$ can be described with the following algorithm (see also Supplementary Note 2 for a more detailed description): first enumerate all inputs using $n$ and map these inputs to their outputs using the map $f$. Then print the resulting list of each output $x$ together with its corresponding probability $P(x)$. If $f$ and $n$ are given in advance, then the algorithmic cost of this operation is $\mathcal{O}(1)$, demonstrating a well-known result from AIT that the algorithmic complexity of a whole set can be much lower than the complexity a typical individual member of the set. Given this set, each output can now be described by an optimal Shannon–Fano–Elias coding[8] with prefix code-words $E(x)$ of length $l(E(x)) = 1 - \log_2 P(x)$ which again can be specified in an $\mathcal{O}(1)$ operation. Since the Kolmogorov complexity of an output $x$ is by definition the shortest algorithm that generates $x$, have provided a bound $K(x|f,n) \leq E(x)$, where $K(x|f,n)$ can be viewed (informally) as the length of computer code required to specify $x$, given that the function $f$ and value $n$ are pre-programmed. Thus, the probability $P(x)$ that a randomly chosen input from $I$, fed into a map $f$, generates an output $x \in O$ can be bounded by:

$$P(x) \leq 2^{-K(x|f,n)+O(1)}. \qquad (2)$$

Note that in contrast to the full AIT coding theorem (1), Eq. (2) only provides an upper bound. It can be viewed as a weaker form of the coding theorem, applicable to computable functions (see also Supplementary Note 2 and refs. [7,10]).

On its own, Eq. (2) may not be that useful, as $K(x|f,n)$ can depend in a complex way on the details of the map $f$ and the input space size $n$. To make progress towards

map independent statements, we restrict the class of maps. The most important restriction is to consider only (1) limited complexity maps for which $K(f) + K(n) \ll K(x) + \mathcal{O}(1)$ in the asymptotic limit of large $x$ (Supplementary Note 3). Using standard inequalities for conditional Kolmogorov complexity, such as $K(x) \le K(x|f, n) + K(f) + K(n) + \mathcal{O}(1)$ and $K(x|f, n) \le K(x) + \mathcal{O}(1)$, it follows for limited complexity maps that $K(x|f, n) \approx K(x) + \mathcal{O}(1)$. Thus, importantly, Eq. (2) becomes asymptotically independent of the map $f$, and only depends on the complexity of the output.

We include three further simple restrictions, namely (2) Redundancy: if $N_I$ and $N_O$ are the number of inputs and outputs respectively then we require $N_I \gg N_O$, so that $P(x)$ can in principle vary significantly, (3) Finite size: we impose $N_O \gg 1$ to avoid finite size effects, and (4) Nonlinearity: We require the map $f$ to be a nonlinear function, as linear transformations of the inputs cannot show bias towards any outputs (Supplementary Note 4). These four conditions are not so onerous. We expect that many real-world maps will naturally satisfy them.

We are still left with the problem that $K(x)$ is formally uncomputable. Nevertheless, in a number of real-world settings, $K(x)$ has been approximated using complexity measures based on standard lossless compression algorithms with surprising success[7]. That is to say, the approximations behave in a manner expected of the true Kolmogorov complexity, and lead to verified predictions. Example settings include: DNA and phylogeny studies[11–13], plagiarism detection[14], clustering music[15], and financial market analysis[16]; see Vitányi[17] for a recent review. These successes suggest that $K(x)$ can in some contexts be usefully approximated[18], even if the exact value of $K(x)$ cannot be calculated. Following the successful applications of AIT above, we assume that the true Kolmogorov complexity $K(x)$ can be approximated by some standard method, such as the ones described above. We will call such an approximation the approximate complexity $\tilde{K}(x)$ to distinguish it from the true Kolmogorov complexity. We therefore need a final condition (5) Well behaved: the map is 'well behaved' in the sense of not producing, for example, a large fraction of pseudorandom outputs such as the digits of $\pi$, which are algorithmically simple, but which have large entropy and thus are likely to have large values of the approximate complexity $\tilde{K}(x)$. For example, pseudorandom number generators are designed to produce outputs that appear to be incompressible, even though their actual algorithmic complexity may be low. Here we mostly use a complexity estimator we call $C_{LZ}(x)$ (Methods section and Supplementary note 7), which is a slightly adapted version of the famous Lempel–Ziv 76 lossless compression measure[19]. But there is nothing fundamental about this choice. In Supplementary Note 7, we show that our main results hold for other complexity measures as well.

Finally, the presence of $\mathcal{O}(1)$ terms is perhaps the least understood limitation for applying formal AIT to real-world settings. Nevertheless, important recent work applying the full AIT coding theorem to very short strings[20,21] has suggested that the presence of $\mathcal{O}(1)$ terms, both in the definition of $K(x)$ and in the coding theorem relationship between $P(x)$ and $K(x)$, does not preclude the possibility of making decent predictions for smaller systems.

Taken together, the arguments above allow us to make our central ansatz, namely that for many real-world maps, the upper bound in Eq. (2) can be approximated as:

$$P(x) \lesssim 2^{-a\tilde{K}(x) - b}, \quad (3)$$

where the constants $a > 0$ and $b$ depend on the mapping, but not on $x$. These constants account for the $\mathcal{O}(1)$ terms and the

particularities of the complexity approximation $\tilde{K}(x)$. Just as for the full coding theorem, there is a strong exponential decay in the probability upper bound upon a linear increase in complexity. This means that high-probability outputs must be simple (have low $\tilde{K}(x)$), while high-complexity outputs must be exponentially less probable. We call such phenomena that arise from Eq. (3) simplicity bias.

In contrast to the full AIT coding theorem, the lack of a lower bound in Eq. (3) means that simple outputs may also have low probabilities. Furthermore (as shown in Supplementary Note 5), we expect the upper bound to decay to a lowest value of about 1/$N_O$ for the largest complexity, $\max(\tilde{K}(x))$. This minimal value for the bound is also the mean probability, and if $P(x)$ is highly biased, many outputs will have probabilities below the mean. In other words, if an output $x$ is chosen uniformly from the set of all outputs, it is not likely to be that close to the bound. On the other hand (Supplementary Note 5) if $x$ is generated by choosing random inputs, which naturally favours outputs with larger $P(x)$, then we can expect that $P(x)$ is relatively close to the bound, at least on a logarithmic scale. In short, the upper bound of the simplicity bias Eq. (3) should be tight for most inputs, but may be weak for many outputs (Of course if the mapping were a full UTM, then Eq. (3) would simply revert to the full coding theorem again with an upper and a lower bound.).

Note also that while simplicity bias means that a simple output $x$ is exponentially more likely to appear when sampling over inputs than a complex output $y$, this does not necessarily mean that simple outputs are more likely than complex outputs because there may be many more of the latter than the former.

In Supplementary Note 8 we show that $a$ can be approximated as:

$$a \approx \frac{\log_2(N_O)}{\max_{x \in O}(\tilde{K}(x))}. \quad (4)$$

It is typically within an order of magnitude of 1. Interestingly, this connection between $a$ and $N_O$ implies that the gradient can be used to predict $N_O$, and vice versa, if an estimate of $\max(\tilde{K}(x))$ can be found, which for some maps can be achieved quite easily. For $b$, we show in Supplementary Note 8 that the default or a priori prediction is $b \approx 0$. Alternatively, access to a relatively small amount of sampled data is usually sufficient to fix $b$. Finally, even if estimating $a$ and $b$ is difficult, Eq. (3) predicts whether $P(y) > P(x)$ or $P(y) < P(x)$ for $y, x \in O$ just using the complexities of $x$ and $y$. In many cases, this prediction is of interest (Supplementary Note 10).

All this raises the question of well the simplicity bias predictions made above hold for real-world maps. The proof of the pudding is in the eating, so we test our predictions for a diverse set of maps described below and shown in Fig. 1.

**Discrete RNA sequence to structure mapping.** One of the best studied discrete input–output maps in biophysics has as inputs RNA nucleotide sequences (genotypes), made from an alphabet of four different nucleotides, and as outputs the RNA secondary structures (SS), phenotypes which specify the bonding pattern of nucleotides[22]. We use the well-known Vienna package[23] that determines the minimum free energy SS for a given sequence (Methods section). Since the binding rules are independent of the length $n$ of input RNA sequences, this is a limited complexity map that satisfies our conditions for simplicity bias (Methods section).

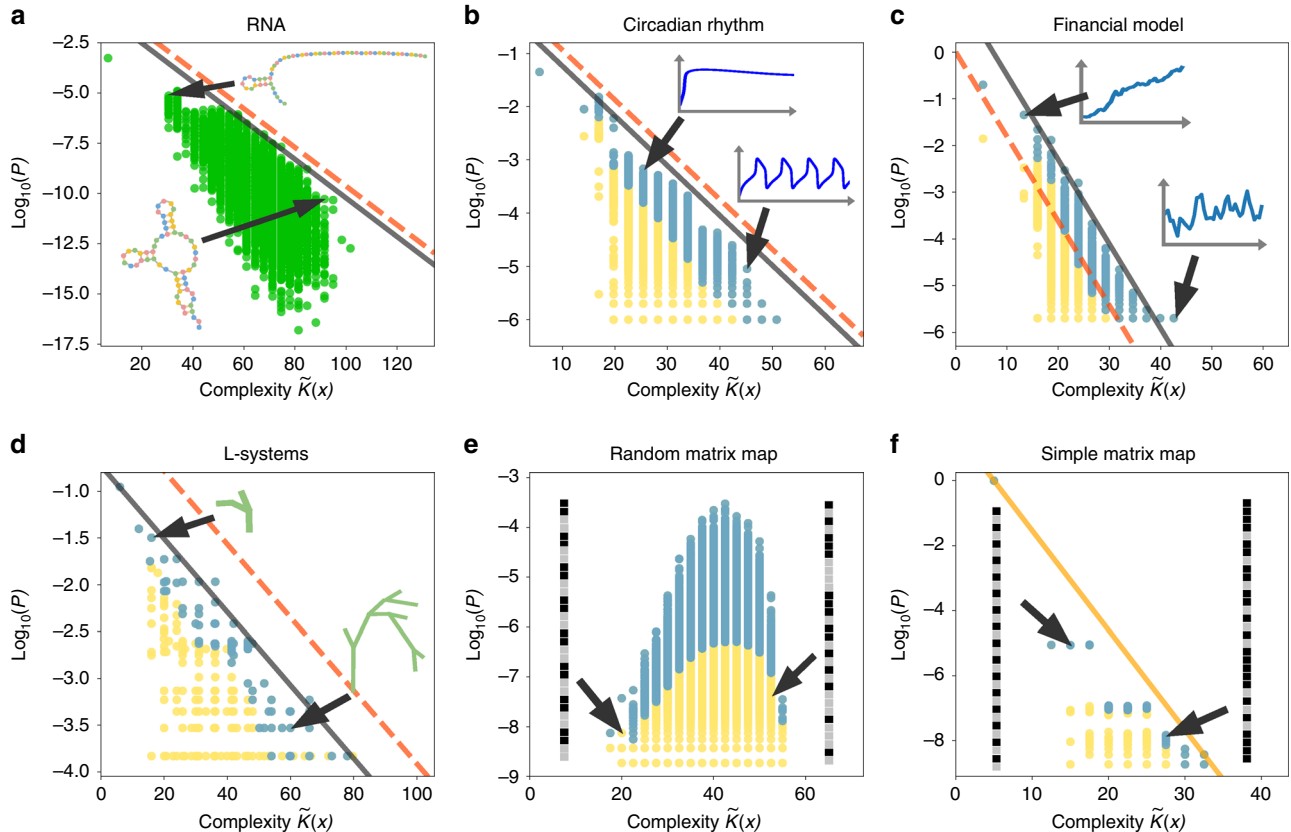

**Fig. 1** Simplicity Bias. The probability $P(x)$ that an output $x$ is generated by random inputs versus the approximate complexity $\tilde{K}(x)$ for **a** the discrete $n = 55$ RNA sequence to SS map (<0.1% of outputs take up 50% of the inputs[24]), **b** the coarse-grained circadian rhythm ODE map (2% of the outputs take up 50% of the inputs), **c** the Ornstein–Uhlenbeck financial model (0.6% of the outputs take up 50% of the inputs), **d** L-systems for plant morphology (3% of the outputs take up 50% of the inputs), **e** a random 32 × 32 matrix map, and **f** a limited complexity 32 × 32 matrix map (both with <0.1% of the outputs taking over 50% of the inputs). Schematic examples of low and high-complexity outputs are also shown for each map. Blue dots are probabilities that take the top 50% of the probability weight for each complexity value while yellow dots denote the bottom 50% of the probability weight (only green was used for **a**, the RNA map, because the output probabilities were calculated using the probability estimator described in ref. [35]). The bold black lines denote the upper bound described in Eq. (3), while the dashed red lines represent the same upper bound, but with the default $b = 0$. For **f**, the upper bound line (orange) was fit to the distribution. All limited complexity maps exhibit simplicity bias, while the random matrix map does not

To estimate the complexity of an RNA SS, we converted a standard dot-bracket representation (Methods section) of the structure into a binary string, and then used our complexity measure $C_{LZ}(x)$ to estimate its complexity. In Fig. 1a, we show $P(x)$ versus $\tilde{K}(x)$ for the $n = 55$ RNA map (Methods section). To compare to Eq. (3), the gradient magnitude $a$ was estimated via Eq. (4) by using previously estimated values of $N_O$[24] together with an estimated value for $\max(\tilde{K})$ (Methods section). To estimate $b$, we used the maximum probability within structures of the modal complexity value (Supplementary Note 8). As can be seen in Fig. 1a, the upper bound prediction of Eq. (3) is remarkably good. All we need to fix its form is some very minimal knowledge of the mapping. Even if we chose the default value $b = 0$, the prediction is still reasonable. This map clearly exhibits the predicted simplicity bias phenomenology.

We further use the RNA map in Supplementary Note 9 to illustrate finite size effects using small system sizes (around $n = 10$) where simplicity bias is much less clear. In Supplementary Note 6, we use $n = 20$ RNA to illustrate an interesting prediction of simplicity bias for inputs: low $\tilde{K}(x)$, low-$P(x)$ outputs, e.g., outputs far from the upper bound, are generated by inputs that have lower than average complexity.

both the input parameters and the outputs. As an example, we take a well-studied circadian rhythm model[25], a system of nine nonlinear ODEs where the inputs are the values of the 15 parameters, and the output is a single-curve $y(t)$ depicting a concentration-time curve for the product at the end of the regulatory cascade (see also Supplementary Note 12). The inputs can be discretised straightforwardly by setting a range and a number of points per range for each parameter (Methods section), while each output curve is (coarsely) discretised to a binary string by using the 'up–down' method[26,27]: for discrete values of $t = \delta t, 2\delta t, 3\delta t \ldots$, if $dy(t)/dt \geq 0$ at position $t = j\delta t$, then a 1 is assigned to position $j$ of the binary string, and otherwise a 0 is assigned (Methods section). The complexity of this map does not change with $n$ and so it can be viewed as a limited complexity map. As can be seen in Fig. 1b, the probability $P(x)$ decays strongly with increasing approximate complexity of the outputs. Again the upper bound (calculated with Eq. (4)) works remarkably well, given the relatively small amount of information needed about the map to fix it. The majority of points generated by random sampling of inputs are within one or two orders of magnitude from the bound. This coarse-grained ODE map shows the same broad simplicity bias phenomenology as the RNA map.

**Coarse-grained ordinary differential equation (ODE).** ODE models can be coarse-grained into discrete maps by discretising

**Ornstein–Uhlenbeck stochastic financial trading model.** In mathematical finance, the Ornstein–Uhlenbeck process, also

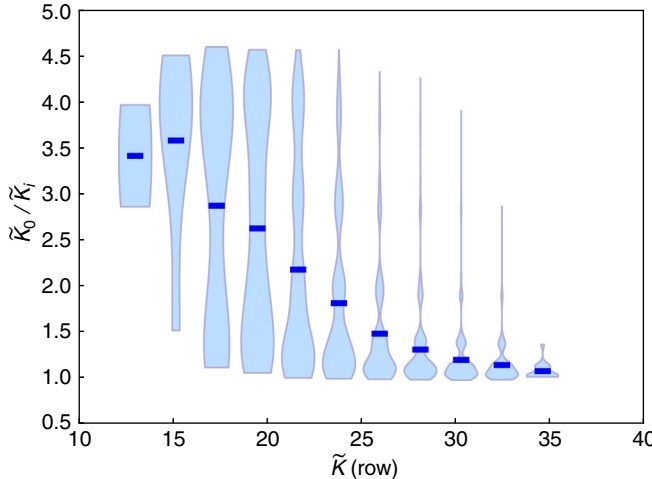

**Fig. 2** Variable complexity matrix map. The complexity of the 20 × 20 circulant matrix map can be varied by changing the complexity $\tilde{K}$(row) of the first row that defines the map. $\tilde{K}_o/\tilde{K}_i$ measures the ratio of the mean complexity of all individual outputs of a given map, divided by the mean complexity of outputs generated by random sampling over all inputs. In this plot, made with $2.5 \times 10^4$ matrices, the distribution of ratios $\tilde{K}_o/\tilde{K}_i$ is shown in a standard violin plot format. The horizontal dark blue lines denote the mean $\tilde{K}_o/\tilde{K}_i$ for each value of $\tilde{K}$(row). Only relatively simple matrix maps (with smaller $\tilde{K}$(row)) exhibit simplicity bias, as indicated by $\tilde{K}_o/\tilde{K}_i$ being significantly >1

known as the Vasicek model[28], is used to model interest rates, currency exchange rates, and commodity prices, and is applied in a trading strategy known as pairs trade[29]. This stochastic process for $S_t$ is governed by $dS_t = \theta(\mu - S_t)dt + \sigma dW_t$, where $\mu$ represents the historical mean value, $\sigma$ denotes the degree of market volatility, $\theta$ is the rate at which noise dissipates, and $W_t$ is a Brownian motion, which is taken as the input to this map. The outputs $x_t$ are sequences over $n$ time steps where $x_j = 0$ if $S_j \leq 0$, and $x_j = 1$ otherwise. The changes between 0 and 1 in the output sequence correspond to changes in the pairs trading strategy: '0' means $S_t$ is below its equilibrium value, so that a trader would profit by buying more of it, while '1' means $S_t$ is above its equilibrium value, and therefore the trader would profit by selling it.

We measure the complexity of these binary output strings using $C_{LZ}(x)$. As shown in Fig. 1c, this map shows basic simplicity bias phenomenology, and the predicted upper bound slope $a$ based on Eq. (4) also works well.

**L-systems**. L-systems[30] are a general modelling framework originally introduced for modelling plant growth, but now also used extensively in computer graphics. They consist of a string of different symbols which constitute production rules for making geometrical shapes. We confined our investigation to non-cyclical graph (i.e., topological tree) outputs. We enumerated all valid L-systems consisting of a single starting letter $F$, followed by rules made of symbols from $\{+, -, F\}$ and length $\leq 9$ (Methods section). The rule set defining the L-systems is independent of input length, and hence this is a limited complexity map. The outputs were coarse-grained to binary strings (Methods section) and again, as can be seen in Fig. 1d, L-systems exhibit simplicity bias. The prediction of the slope $a$ based on Eq. (4) again works well.

**Random matrix map with bias but not simplicity bias**. Finally, we provide an example of a map that exhibits strong bias that is not simplicity bias. We define the matrix map by having binary input vectors $p$ of length $n$ that map to binary output vectors $x$ of

same length though $x_i = \Theta((M \cdot p)_i)$, where $M$ is a matrix and the Heaviside thresholding function $\Theta(y) = 0$ if $y < 0$ and $\Theta(y) = 1$ if $y \geq 0$. This last nonlinear step, which resembles the thresholding function in simple neural networks, is important since linear maps do not show simplicity bias. In Fig. 1e, we illustrate this map for a matrix made with entries randomly chosen to be −1 or +1. The rank plot (Supplementary Fig. 17) shows strong bias, but in marked contrast to the other maps, the probability $P(x)$ does not correlate with the complexity of the outputs. Simplicity bias does not occur because the $n \times n$ independent matrix elements mean that the mapping's complexity grows rapidly with increasing $n$ so that the map violates our limited complexity condition (1). Intuitively: the pattern of 1s and 0s in output $x$ is strongly determined by the particular details of the map $M$, and so does not correlate with the complexity of the output.

To explore in more detail how simplicity bias develops or disappears with limited complexity condition (1), we also constructed a circulant matrix where we can systematically vary the complexity of the map. It starts with a row of $p$ positive 1s and $q - 1s$, randomly placed. The next row has the same sequence, but permuted by one. This permutation process is repeated to create a square $n \times n$ matrix. Thus the map is completely specified by defining the first row together with the procedure to fill out the rest of the matrix. In Fig. 2, we plot the ratio of the mean complexity sampled over outputs ($\tilde{K}_o$) divided by the mean complexity sampled over all inputs ($\tilde{K}_i$), as a function of the complexity of the first row that that defines the matrix, $\tilde{K}$(row). If the ratio $\tilde{K}_o/\tilde{K}_i$ is significantly larger than one, then the map shows simplicity bias. Simplicity bias only occurs when $\tilde{K}$(row) is very small, i.e., for relatively simple maps that respect condition (1). The output of one such simple matrix maps is shown in Fig. 1f. In Supplementary Note 12, we investigate these trends in more detail as a function of matrix type, size $n$, and also investigate the role of matrix rank and sparsity.

## Discussion

While the full AIT coding theorem has been established only in the abstract and idealised setting of UTMs and uncomputable complexities, nonetheless a general inverse relation between complexity and probability is intuitively reasonable. We recast a weaker version of the coding theorem for computable maps into the practical form of Eq. (3) for limited complexity maps. The basic simplicity bias phenomenology this equation predicts holds for a wide diversity of real-world maps as demonstrated in Fig. 1 with further maps shown in Supplementary Note 12.

Nevertheless, many questions remain. For example, our derivations typically suffer from $\mathcal{O}(1)$ terms that are hard to explicitly bound except in the limit of large $x$. So it is perhaps surprising that Eq. (3) works so well even for smaller systems that are not in this limit. We conjecture that, just as is found elsewhere in science and engineering, our results for asymptotically large $x$ approximately apply well outside the domain where they can be proven to hold.

We argue that the bound (3) should be valid for limited complexity maps, and provide as a counter-example a random matrix map that shows bias which is not simplicity bias. We can also vary the complexity of this matrix map by making it simpler so that simplicity bias emerges. Nevertheless, a more general theory of how and when this crossover to simplicity bias occurs as a function of map complexity still needs to be worked out.

Kolmogorov complexity is technically uncomputable and we approximate it with a compression-based measure. There may be classes of maps, such as pseudorandom number generators, for which such approximations breakdown. Also, while it is true that most outputs generated by random sampling of inputs are likely

to be close to our upper bound, in contrast to the original AIT coding theorem for UTMs which has an upper and lower bound which are asymptotically close, our computable maps have many outputs that fall well below the upper bound. Understanding why these particular outputs fall far below the bound is an important topic of future investigation.

One way to answer some of these questions may be to systematically investigate a hierarchy[31] of more abstract machines with less computational power than a UTM—ranging from simple finite state transducers[32] to context free grammars (e.g., RNA[33]) to more complex context sensitive grammars—and so to search for more general principles. A completely different direction for investigation may be to study individual maps in much more detail. For some simpler maps (see e.g., the random walk and polynomial examples in Supplementary Note [12]), explicit probability-complexity relations that resemble the bound of Eq. ([3]) could be derived using, for example, well established links between Shannon entropy and Kolmogorov complexity[7].

In this paper, we have focussed on maps that satisfy a number of restrictions. In particular, it will be interesting to study maps that violate our condition (5) of being well behaved. In parallel, another interesting open question remains: how much of the patterns that we study in science and engineering are governed by maps that that are simple and well-behaved. Another possible future research direction would be to explore connections to leaning theory, including links to the minimum description length (MDL) principle first introduced by Rissanen[34], which has been applied in the context of statistical inference and data compression. Similar to our work here, MDL theory has been advanced in an attempt to apply ideas from AIT to practical concrete problems, and thus there may be more connections to explore.

Finally, our prediction of an exponential decay in probability with a linear increase in complexity is strong and general. We expect many different applications of simplicity bias across science and engineering. Working out the implications for individual systems will be an important future task.

## Methods

**Complexity estimator**. We approximate the complexity of a binary string $x = \{x_1 \ldots x_n\}$ as

$$C_{LZ}(x) = \begin{cases} \log_2(n), & x = 0^n \text{ or } 1^n \\ \log_2(n) \frac{1}{2} [N_w(x_1 \ldots x_n) + N_w(x_n \ldots x_1)] & \text{otherwise} \end{cases}, \quad (5)$$

where $n = |x|$ and $N_w(x)$ is the number of words (distinct patterns) in the dictionary created by Lempel–Ziv algorithm[19]. The reason for distinguishing $0^n$ and $1^n$ is to correct an artefact of $N_w(x)$, which assigns complexity 1 to the string 0 or 1 but complexity $2-0^n$ or $1^n$ for $n \geq 2$. The Kolmogorov complexity of such a trivial string scales as $\log_2(n)$ as one only needs to encode $n$. In this way, we ensure that our $\tilde{K}(x) = C_{LZ}(x)$ measure not only gives the correct behaviour for complex strings in the $\lim_{n \to \infty}$, but also the correct behaviour for the simplest strings. Taking the mean of the complexity of the forward and reversed string makes the measure more fine grained in the sense of having more different possible complexity values. We discuss this complexity estimator in more detail in Supplementary Note 7.

**RNA secondary structure**. This map is determined by basic physiochemical laws, and does not grow with $n$. Furthermore, the number of inputs (sequences), which grows as $4^n$, is much larger than the number of relevant secondary structures[22,24], and so $N_I \gg N_O$. For large enough $n$, where finite size effects are no longer important (Supplementary Note 9), this system satisfies our conditions for simplicity bias. In our analysis, folding RNA sequences to secondary structures was performed using the Vienna package[23] with all parameters set to their default values (e.g., the temperature $T = 37\,°C$). Total of 20,000 random RNA sequences were generated, then folded. Due to the large size of this system ($4^{55} \approx 1.3 \times 10^{33}$ inputs and $\sim 10^{13}$ outputs[24]), it is impractical to determine probabilities by sampling and counting frequencies of output occurrence. Instead, to determine $P(x)$ for each sampled structure, we used the neutral network size estimator (NNSE) described in ref. [35], which employs sampling techniques together with the inverse fold algorithm from the Vienna package. We used default settings except for the total number of measurements (set with the -m option) which we set to 1 instead of

the default 10, for the sake of speed. More details of our methods can also be found in ref. [24]. By random sampling, we likely have only reached a small fraction of all the outputs, so it was not possible in Fig. 1a to calculate what the top 50% of outputs were. But in ref. [24] we calculate that for this length, only 0.1% of outputs take up over 50% of inputs, so the map is highly biased with most inputs mapping to outputs relatively close to the upper bound.

To estimate the complexity of an RNA SS, we first converted the dot-bracket representation of the structure into a binary string $x$, and then used $C_{LZ}(x)$ to estimate its complexity. To convert to binary strings, we replaced each dot with the bits 00, each left-bracket with the bits 10, and each right-bracket with 01. Thus an RNA SS of length $n$ becomes a bit string of length $2n$. As an example, the following $n = 12$ structure yields the displayed 24-bit string

$$(((( \ldots ))) \ldots \; \rightarrow \; 101010000000010101000000.$$

The gradient magnitude $a = 0.32$ in our upper bound was estimated via Eq. ([4]) by using our estimated values of $N_O$ from ref. [24], in addition to an estimated value for $\max(\tilde{K})$. For the latter quantity, we made the approximation that $\max(\tilde{K}) = C_{LZ}(\zeta_{2n})$, where $\zeta_{2n}$ is a random bit string of length $2n$ made up of randomly choosing $n$ pairs 00, 10 and 01. This choice of randomisation is due to observing that a first order approximation to a random RNA structure is a uniform sampling of dot, left-bracket, right-bracket, which we them write in binary, as described. We took the largest complexity over 250 random bit string samples. To estimate $b = 3.2$, we used the sampled data to find the maximum probability within outputs with complexity equal to the mode complexity value (Supplementary Note 8).

We also note that the random sampling of genotypes will strongly favour outputs with larger $P(x)$. We have therefore only sampled a tiny fraction of the whole space of outputs for $n = 55$ RNA (Supplementary Fig. 5 and ref. [24]). There are likely many more low-probability outputs, even at the lower complexities. However, the overall probability of generating these outputs will be low[24], and so should not affect our main conclusions.

**Coarse-grained ODE**. For the ODE system used in the circadian rhythm map, we set the possible input values for each of the 15 parameters by multiplying the original value[25] by one of {0.25, 0.50, …, 1.75, 2.00}, chosen with uniform probability. The total number of inputs is then $N_I = 8^{15} \approx 3 \times 10^{13}$. In Supplementary Note 12, we show that this choice of input discretisation and sample size and initial conditions does not qualitatively affect our results.

To generate the plot in Fig. 1b, $10^6$ inputs were sampled, and outputs were discretised with $\delta t = 1$ and $t \in [1, 50]$, thus producing a 50-bit output string from the concentration of the product at the end of the regulatory cascade over time. The slope $a = 0.31$ was obtained via Eq. ([4]), using the values of $N_O$ and $\max(\tilde{K})$ from the full enumeration of inputs, while $b = 1.0$ was fit to the distribution.

**Ornstein–Uhlenbeck financial model**. For the Ornstein–Uhlenbeck model presented in Fig. 1c, the parameters were $S_0 = 1$, $\theta = 0.5$, $\mu = 0.5$ and $\mu = 0$, and 40 time steps. $10^6$ samples were made, and given that the Brownian motion $dW_t$ allows for steps of any size (even at a low probability), the possible outputs $O$ are all $2^{40}$ binary strings of length 40, making it possible to calculate $\max(\tilde{K}(x))$ and $N_O$. The slope $a = 0.60$ was obtained via Eq. ([4]), and $b = -4.38$ was obtained using the modal complexity value. In Supplementary Note 12, we show that the same behaviour obtains for different parameter combinations, and we show that under certain conditions, the model reduces to a simpler random walk return problem treated in Supplementary Note 12 for which simplicity bias is also obtained.

**L-systems**. All valid L-systems consisting of a single starting letter $F$, followed by rules made of symbols from $\{+, -, F\}$ and length $\leq 9$ were generated. The outputs were coarse-grained using a method suggested in ref. [36] to associate binary strings to any non-cyclical graph with a distinguished node called the 'root' (these graphs are known as rooted trees). Specifically, by walking along the branches, starting right, and recording whether it is going up (0) or down (1), a binary string representation of the tree is made. The slope $a = 0.13$ was obtained via Eq. ([4]), using the values of $N_O$ and $\max(\tilde{K})$ from the full enumeration of inputs, while $b = 2.41$ was obtained using the modal complexity value, as above.

**Matrix map**. For the matrix map represented in Fig. 1f, we took a $32 \times 32$ matrix with all entries chosen uniformly from $\{-1, 1\}$, and sampled $10^9$ out of $2^{32} \approx 4 \times 10^9$ inputs. Further examples of this random map can be found in Supplementary Note 12. For the circulant matrices used in Figs. 1e and 2, we generated the first row, which determines the map, with entries chosen uniformly from $\{-1, 1\}$. For the simple map in Fig. 1e, we chose a first row with a low complexity $\tilde{K}(\text{row})$. In this map the estimate of the slope from Eq. ([4]) did not work well possibly because a large fraction of the inputs map to a single-output vector made up of all 0s. So in Fig. 1e, the slope was simply fit to the data. Further discussion of the circulant matrix map can be found in Supplementary Note 12.

**Data availability**. The data sets generated during and/or analysed during the current study are available from the corresponding authors on reasonable request.

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

## Acknowledgements

We thank B. Frot for providing data for the L-systems and S.E. Ahnert, B. Frot, P. Gács, I.G. Johnston and H. Zenil for helpful discussions. We thank the EPSRC for funding K.D. through EPSRC/EP/G03706X/1 and the Clarendon Fund for funding C.Q.C.

## Author contributions

A.A.L. and K.D. conceived the study. K.D., and C.Q.C. performed the computational analyses. A.A.L., and K.D. performed the theoretical analyses. A.A.L., K.D., and C.Q.C wrote the paper.
