## [Peer Review File · Nature Communications]

Reviewers' comments:

Reviewer #1 (Remarks to the Author):

By extending a classical result from algorithmic information theory, the major claim of the paper is that for many real-world maps, the a priori probability $P(x)$ that randomly sampled inputs generate a particular output x decays exponentially with the approximate Kolmogorov complexity $K(x)$ of that output.

This is a surprising and somewhat important general result. It concerns many fields as the paper shows.

Their weaker version of the coding theorem for computable maps into a simple equation is really practically important and the paper shows that the basic simplicity bias phenomenology this equation predicts holds for a wide diversity of real-world

This is an original paper.

The statistical analysis is rather elementary and convincing.

Line 91 "It is not hard to show (see SI section II), using using standard results from AIT such as the complexity of a whole set can be much lower than the complexity of individual members of the set, that the probability $P(x)$ that a randomly ... "

The word using is written twice

More important: this is a central point of the paper, and it must be explained more cautiously.

The paper must be accepted after some corrections (I think the English can be improved) and complements about the central equation of the paper.

Reviewer #2 (Remarks to the Author):

In this paper it is uncovered that the a priori probability $P(x)$ that randomly sampled inputs of different maps generate simple outputs decays exponentially with their complexity (the same measured by the approximate Kolmogorov complexity). The authors call this behavior simplicity bias.

Based on information theory, an upper bound for most inputs is derived. The described behavior is investigated in different real systems.

Some concern about the paper is related to the restrictions applied to the maps, particularly the necessity of dealing with low complexity maps (as showed by the authors, high complexity maps, as in the random matrix case, do not exhibit simplicity bias), and the condition of “well behaved” which seems too restrictive for real nonlinear maps. For example, maps that mostly produce chaotic solutions (as should be the case for some nonlinear partial differential equations) do not shown simplicity bias?

Moreover, regarding the practical usefulness of the results, it is interesting that, in some cases, low complexity outputs of low probability are produced by low complexity inputs. However, there is not a lower bound and, thus, it is possible that particular inputs can also produce high complexity outputs of low probability. The question arises, thus, if such high complexity outputs of low probability should be generated by low complexity inputs. Such possibility is interesting for predicting the a priori probability of successful modelling of complex behaviors with simple maps feed by simple inputs. For example, in tRNA folding, a very interesting question regards the relation between the complexity of the input sequences and the complexity of the folded tRNAs. The probability that a random input sequence generates a complex folding like such observed in present tRNA molecules is relevant from an evolutionary point of view.

Another point regards the example of coarse grained ODEs as a model for circadian rhythms. The behaviors showed in the cases of low and high complexity are respectively, a fixed point and a self-sustained oscillation. However, we known that such behaviors depend also strongly on initial conditions; thus, also a randomization of initial conditions seems to be necessary for asserting a global trend in the probability of the output complexity (the simplicity bias).

Nevertheless, the paper is clear, scientifically sound, address an interesting issue, and show new results that can stimulate further research in this topic.

In my opinion the paper deserve publication in the Journal as it is, perhaps a little discussion about the influence in practical terms of the requirements on complexity and well behavior of the map should increase its reach.

Reviewer #3 (Remarks to the Author):

Professor and Chair

School of Computer Science

Georgia Institute of Technology

Reviewers' comments:

Reviewer #1 (Remarks to the Author):

By extending a classical result from algorithmic information theory, the major claim of the paper is that for many real-world maps, the a priori probability $P(x)$ that randomly sampled inputs generate a particular output x decays exponentially with the approximate Kolmogorov complexity $\tilde{K}(x)$ of that output.

This is a surprising and somewhat important general result. It concerns many fields as the paper shows.

Their weaker version of the coding theorem for computable maps into a simple equation is really practically important and the paper shows that the basic simplicity bias phenomenology this equation predicts holds for a wide diversity of real-world

This is an original paper.

The statistical analysis is rather elementary and convincing.

Line 91 "It is not hard to show (see SI section II), using using standard results from AIT such as the complexity of a whole set can be much lower than the complexity of individual members of the set, that the probability $P(x)$ that a randomly ... "

The word using is written twice

We thank the referee for pointing this error out, it is now corrected.

More important: this is a central point of the paper, and it must be explained more cautiously.

This argument is laid out in much more detail in the SI section II.

Nevertheless, we have taken this comment on board and rewritten and

expanded the main text around what was line 91 to sketch out the main argument.

The paper must be accepted after some corrections (I think the English can be improved) and complements about the central equation of the paper. We have had a careful read through the paper and have done our best to improve the English where we could.

Reviewer #2 (Remarks to the Author):

In this paper it is uncovered that the a priori probability $P(x)$ that randomly sampled inputs of different maps generate simple outputs decays exponentially with their complexity (the same measured by the approximate Kolmogorov complexity). The authors call this behavior simplicity bias. Based on information theory, an upper bound for most inputs is derived. The described behavior is investigated in different real systems.

Some concern about the paper is related to the restrictions applied to the maps, particularly the necessity of dealing with low complexity maps (as showed by the authors, high complexity maps, as in the random matrix case, do not exhibit simplicity bias), and the condition of "well behaved" which seems too restrictive for real nonlinear maps. For example, maps that mostly produce chaotic solutions (as should be the case for some nonlinear partial differential equations) do not shown simplicity bias?

It this paper we did not treat maps that are chaotic, or related maps such as pseudo-random number generators, all which are violate condition (5) on page 2 of the main paper i.e. they are not "well behaved" because they produce pseudorandom numbers. As explained in the main text, the main reason for imposing this condition is that these pseudorandom high entropy outputs would be mistakenly classified by our Lempel Ziv (LZ) estimator or almost any other known complexity estimator, as having high complexity. Pseudorandom number generators are in fact designed to produce numbers that are hard to compress (i.e. they fool most complexity estimators), even if their origin is algorithmic, so that their true Kolmogorov complexity can be quite small. We did indeed study some simple chaotic systems such as the logistic map and found intriguing results suggesting that some aspects of simplicity bias may still hold. However, we feel that the complexities mentioned above mean that more needs to be worked out before this could be published. We believe that studying simplicity bias phenomenology for maps that are not "well behaved" is indeed a promising new avenue of research, as the referee implies, and now mention this idea explicitly in the discussion section of the paper.

Moreover, regarding the practical usefulness of the results, it is interesting that, in some cases, low complexity outputs of low probability are produced

by low complexity inputs. However, there is not a lower bound and, thus, it is possible that particular inputs can also produce high complexity outputs of low probability. The question arises, thus, if such high complexity outputs of low probability should be generated by low complexity inputs. Such possibility is interesting for predicting the a priori probability of successful modelling of complex behaviors with simple maps feed by simple inputs. For example, in tRNA folding, a very interesting question regards the relation between the complexity of the input sequences and the complexity of the folded tRNAs. The probability that a random input sequence generates a complex folding like such observed in present tRNA molecules is relevant from an evolutionary point of view.

The referee again makes an excellent point about the possible evolutionary implications of our work. We have written on RNA evolution previously (e.g. our reference [20]) and are currently preparing a more general paper on the evolutionary implications of simplicity bias for publication.

Another point regards the example of coarse grained ODEs as a model for circadian rhythms. The behaviors showed in the cases of low and high complexity are respectively, a fixed point and a self-sustained oscillation. However, we known that such behaviors depend also strongly on initial conditions; thus, also a randomization of initial conditions seems to be necessary for asserting a global trend in the probability of the output complexity (the simplicity bias).

To address this point, we took a sample of 10^6 combinations of different values for parameters and initial conditions, produced their corresponding outputs and derived the upper bound coefficients for the slope $a=0.32$ and offset $b=1.39$ according the methods proposed in the text. The result, shown in Figure 8b in the SI, shows the same simplicity bias as in Figure 1b in the main paper.

Nevertheless, the paper is clear, scientifically sound, address an interesting issue, and show new results that can stimulate further research in this topic.

In my opinion the paper deserve publication in the Journal as it is, perhaps a little discussion about the influence in practical terms of the requirements on complexity and well behavior of the map should increase its reach.

We thank the referee for suggesting this. At the end of the discussion, we have now added the following text:

"In this paper, we have focussed on maps that satisfy a number of restrictions. In particular, it will be interesting to study maps that violate our condition (5) of being well-behaved.

In parallel, another interesting open question remains: how much of the patterns that we study in science and engineering are governed by maps that are simple and well-behaved?"

Reviewer #3 (Remarks to the Author):

To put these comments into context, my research expertise lies in computational complexity and I have done extensive work in Kolmogorov complexity, or in the author's terminology algorithmic information theory.

The basic idea of algorithmic information is that you can describe the "algorithmic information" of some data by the size of the shortest program that generates that data. So a sequence "ababababababababababababababab" has less information than "abaababaaabaabbbababbbabaababa" even though they would have the same probability if chosen randomly. Dual to this notion is the universal measure $p(x)$ that weights data x by $2^{-K(x)}$ where $K(x)$ is the algorithmic information complexity of x .

This paper looks at functions that take uniformly randomly chosen inputs and run them through some "natural" process and look at the probability generated by the output strings. If the process simulated a arbitrary computation then it is known that the probability distribution would be within a constant factor of $p(x)$. The authors do some computational experiments to show that you can see this kind of behavior to processes such as structure from RNA sequence, circadian rhythm and a financial market model. To simulate the uncomputable Kolmogorov complexity, the authors use standard compression functions, something that has been used with some success in the past.

For the most part, the author's discussion of technicalities of Kolmogorov complexity pretty accurate but I found the interpretation of the results here quite simplistic. Equation (3) just gives an upper bound, technically even the random matrix map has this bound, albeit with a large a and b .

Indeed, in a trivial sense, $P(x) < 1$, so an upper bound always holds. But for the matrix map, for example, the complexity of the outputs does not correlate in an obvious way with the probability, and it certainly does not decrease with complexity, as would be needed for any slope $a > 0$ large or small. So, equation (3) only holds in this kind of trivial sense if you set the slope a to zero and just use the offset b .

And since the authors use standard compression functions instead of true Kolmogorov complexity, really what is going on is the ability of the

compression function to sort of “undo” the functions given. If you use a cryptographic hash function (like MD5 or SHA-2), you should find similar behavior to the random matrix map even though these hash functions have small descriptions.

As described in our response to referee 2, we restrict ourselves to maps that don't violate condition (5) on page 2. Pseudorandom number generators are a good example of maps that do violate this condition. They are precisely designed to do so. We have now added a sentence on the page where we introduce condition (5) explaining this explicitly for pseudorandom number generators on page 2. It reads:

“For example, pseudorandom number generators are designed to produce outputs that appear to be incompressible, even though their actual algorithmic complexity may be low.”

As we demonstrate with our broad range of examples, this restriction does not preclude a large number of interesting possible applications of simplicity bias. It would be interesting to study systems that do violate condition (5), and we now added the following text to discussion at the end of the paper

“` In this paper, we have focussed on maps that satisfy a number of restrictions. In particular, it will be interesting to study maps that violate our condition (5) of being well-behaved.

In parallel, another interesting open question remains: how much of the patterns that we study in science and engineering are governed by maps that that are simple and well-behaved? `”

The authors do not make a great argument for the motivation for a simplicity bias.

We are not quite sure what the reviewer means here. It could be that the reviewer means: Why care about simplicity bias? We argue here that a large number of physically relevant maps show a bias towards simple outputs, and that we can provide a useful bound on the probabilities using concepts from AIT. Our results have broad implications across science and engineering, and that seems to us like motivation enough.

Or perhaps the reviewer means: Can you prove that simplicity bias will arise? We address this interesting question in SI section IV, where we show that we can construct maps that are simple, and yet do not show simplicity bias. Admittedly, these maps are somewhat artefactual, but they demonstrate that it is not so easy to prove exactly when simplicity bias will

or will not occur, even if we observe that it is widespread. We leave this interesting question for future work.

Is it really just capturing the phenomenon that outputs of a simple function with a large number of inverses occur with higher probability and can often be compressed because of it?

In compression theory, outputs can be compressed to $\sim \log(1/P(x))$ bits, and hence high probability outputs (with many inverses) can be assigned short code words. In this sense, outputs with many inverses can be compressed, because of this fact, regardless of their nature (what the outputs look like). This is a standard result from Shannon information theory, and it doesn't really matter what the objects are, the coding has to do with their distribution. However, as we briefly discuss in SI IB, Kolmogorov complexity is not about distributions, as Shannon information is, but rather about individual objects. So, it is not at all obvious in the cases we examine that higher probability outputs should *themselves* be simple/compressible, i.e. why should a high probability output be symmetric/regular, and hence compressible? We find that they are compressible due to inherent regularities, not simply due to having a high probability. The fact that we can connect their probability to their complexity is the central claim of this paper.

I think there might be a better story to tell in connections to learning and predictions along the lines described in Li & Vitanyi (3rd edition), section 5.4.

We agree with the referee that there is an interesting story connecting our paper to learning and prediction. In fact, we are currently working hard on a paper that makes connections to learning theory.

We also are aware of section 5.4 of Li & Vitanyi's book, which summarises work around the Minimum Description Length (MDL), introduced by Rissanen in 1978. Roughly speaking, MDL says that given some data, the hypothesis H we choose to explain it should minimise both the complexity of the hypothesis $K(H)$ and the complexity of the data D , given the hypothesis. That is, we should seek to minimise $K(H)+K(D|H)$. While MDL is relevant to learning problems, it is not yet clear whether MDL can shed light on simplicity bias. Just to be clear: One of the main points of our paper is to say that given a set of outputs from a simple map, we can make non-trivial predictions about which outputs will have higher/lower

probability, just by measuring (or estimating) the complexity of each output. Hence in this paper, we are not trying to form hypotheses or learn from observed data, rather we are making a priori predictions without/before looking at the data.

Nevertheless, the connections to MDL are a potentially interesting research topic for future work, and we have now included a line in the discussion that cites MDL and mentions possible connections.

Lance Fortnow
Professor and Chair
School of Computer Science
Georgia Institute of Technology

REVIEWERS' COMMENTS:

Reviewer #1 (Remarks to the Author):

For me the new version of the paper is good for publication.

Reviewer #2 (Remarks to the Author):

The authors have answered satisfactorily all raised concerns and have made an effort for including some suggested comments.

Thus, in my opinion, the new version of the paper deserves publication in Nature Communications as it is.

Reviewer #3 (Remarks to the Author):

In their rebuttal the authors state: "However, as we briefly discuss in SI IB, Kolmogorov complexity is not about distributions, as Shannon information is, but rather about individual objects. So, it is not at all obvious in the cases we examine that higher probability outputs should themselves be simple/compressible, i.e. why should a high probability output be symmetric/regular, and hence compressible? We find that they are compressible due to inherent regularities, not simply due to having a high probability. The fact that we can connect their probability to their complexity is the central claim of this paper."

This misses the real difference between Kolmogorov and Shannon. You can define a distribution $m(x) = 2^{-K(x)}$, a universal distribution. In fact Kolmogorov originally started with the distribution and then took logs to get a measure on strings. The difference with Shannon is that Kolmogorov

takes into account computation. That's why equation (3) is not surprising and actually tight with real Kolmogorov complexity.

My point with Equation (3) is that you only have an upper bound, not a lower bound. When you aren't tight it's not because of the Kolmogorov complexity but because you are using a weak version of it.

RESPONSE TO REFEREES.

Referees 1 and 2 make no further comments, and referee 3 makes some small points that we answer in red below.

Referee 3 writes:

In their rebuttal the authors state: “However, as we briefly discuss in SI IB, Kolmogorov complexity is not about distributions, as Shannon information is, but rather about individual objects. So, it is not at all obvious in the cases we examine that higher probability outputs should themselves be simple/compressible, i.e. why should a high probability output be symmetric/regular, and hence compressible? We find that they are compressible due to inherent regularities, not simply due to having a high probability. The fact that we can connect their probability to their complexity is the central claim of this paper.”

This misses the real difference between Kolmogorov and Shannon. You can define a distribution $m(x) = 2^{-K(x)}$, a universal distribution. In fact Kolmogorov originally started with the distribution and then took logs to get a measure on strings. The difference with Shannon is that Kolmogorov takes into account computation. That’s why equation (3) is not surprising and actually tight with real Kolmogorov complexity.

We agree that the key difference is that Kolmogorov takes into account computation. In the Supplementary Information section IB, we briefly discuss Shannon v.s. Kolmogorov and also refer the reader to standard works that discuss this difference further. We feel there is no need to discuss this particular question further as much more detailed treatments are found in the standard literature, and this is not a major point of our paper.. Equation (3) is indeed tight with real Kolmogorov complexity and a universal Turing machine, when it reverts to the coding theorem of equation (1). However here we are looking at systems that are not UTMs, which is the main point of our paper. So we are essentially on the same page as the referee here. To make this more clear, we have now added, in the paragraph right after eq(3), a line that reads:

(Of course if the mapping were a full UTM, then equation~(\ref{eq:approx}) would simply revert to the full coding theorem again with an upper and a lower bound.)

My point with Equation (3) is that you only have an upper bound, not a lower bound. When you aren’t tight it’s not because of the Kolmogorov complexity but because you are using a weak version of it.

We agree, and to make this point more clear, we have now added a sentence to the end of the introduction which reads:

In the next sections, we show how a weaker version of the coding theorem, which approximately preserves the exponential preference for low complexity, can be applied to a wide range of practical systems.